# Nrf2 Pathway and Oxidative Stress as a Common Target for Treatment of Diabetes and Its Comorbidities

**DOI:** 10.3390/ijms25020821

**Published:** 2024-01-09

**Authors:** Michelle Yi, Leslie Cruz Cisneros, Eric J. Cho, Michael Alexander, Francesca A. Kimelman, Lourdes Swentek, Antoney Ferrey, Ekamol Tantisattamo, Hirohito Ichii

**Affiliations:** 1Department of Surgery, University of California Irvine, Irvine, CA 92697, USA; myi10@hs.uci.edu (M.Y.); lcruzcis@hs.uci.edu (L.C.C.); ejcho5@hs.uci.edu (E.J.C.); michaela@hs.uci.edu (M.A.); fkimelma@hs.uci.edu (F.A.K.); lyrobles@hs.uci.edu (L.S.); 2Department of Medicine, University of California Irvine, Irvine, CA 92697, USA; ferreya@hs.uci.edu (A.F.); etantisa@hs.uci.edu (E.T.)

**Keywords:** diabetes, heart, kidney, liver, oxidative stress, Nrf2

## Abstract

Diabetes is a chronic disease that induces many comorbidities, including cardiovascular disease, nephropathy, and liver damage. Many mechanisms have been suggested as to how diabetes leads to these comorbidities, of which increased oxidative stress in diabetic patients has been strongly implicated. Limited knowledge of antioxidative antidiabetic drugs and substances that can address diabetic comorbidities through the nuclear factor erythroid 2–related factor 2 (Nrf2) pathway calls for detailed investigation. This review will describe how diabetes increases oxidative stress, the general impact of that oxidative stress, and how oxidative stress primarily contributes to diabetic comorbidities. It will also address how treatments for diabetes, especially focusing on their effects on the Nrf2 antioxidative pathway, have been shown to similarly affect the Nrf2 pathway of the heart, kidney, and liver systems. This review demonstrates that the Nrf2 pathway is a common pathogenic component of diabetes and its associated comorbidities, potentially identifying this pathway as a target to guide future treatments.

## 1. Introduction

Diabetes is a chronic disease characterized by the dysregulation of glucose metabolism, leading to hyperglycemia. This hyperglycemia is caused by the autoimmune destruction of beta-cells, resulting in insulin insufficiency, or insulin resistance. As diabetes contributes to the incidence of macrovascular complications commonly involving cardiac and cerebrovascular disease and peripheral arterial disease, as well as microvascular complications causing retinopathy, neuropathy, nephropathy, and more, it is ranked as the seventh leading cause of death in the United States [1].

There are several theories on how hyperglycemia leads to dysfunction in other organs. One classic theory is that chronic hyperglycemia in the vascular supply of organs and peripheral tissues leads to an increased production of advanced glycation end-products (AGEs). AGEs are proteins or lipids that are glycated because of their exposure to glucose. The build-up of AGEs can induce thickening of the capillary basement membrane, in addition to other microvascular complications [2]. AGEs can promote oxidative stress, inflammation, and apoptosis of the affected tissues [3]. The thickening of microvascular structure, particularly the capillary basement membrane of the glomeruli, retina, myocardium, skin, and muscle, is referred to as diabetic microangiopathy. Chronic diabetic microangiopathy has been associated with hypertension and tissue hypoxia [4], inducing damage in the associated tissue. Diabetes-related hypertension is a strong risk factor for diabetic nephropathy [5,6]. While there are a substantial number of studies performed on various substances that can treat both diabetes and its comorbidities [7,8,9,10,11], there is limited knowledge of common antidiabetic drugs and other antidiabetic substances that can address various diabetic comorbidities through the nuclear factor erythroid 2-related factor 2 (Nrf2) pathway.

This review will discuss how diabetes leads to increased oxidative stress through various mechanisms, and how oxidative stress has been linked to several diabetic comorbidities. It will also describe specific cardiovascular, renal, and hepatic comorbidities associated with diabetes and how treatments for diabetes may directly affect the recovery of the organ systems harmed by such comorbidities.

## 2. How Diabetes Induces Oxidative Stress

Several mechanisms have been suggested to explain reactive oxygen species (ROS) production in hyperglycemic environments.

First, autoxidation of glucose becomes a source of ROS. Glucose is oxidized into enediol radical anion, which is converted to reactive keto-aldehydes, and then to superoxide anion radicals. This pathway also produces hydrogen peroxide, which requires degradation by glutathione peroxidase or catalase [12].

Second, the polyol pathway or aldose reductase pathway implicates the conversion of glucose to sorbitol in the setting of hyperglycemia as a source of oxidative stress [13]. In this pathway, sorbitol levels increase due to glucose metabolism by aldose reductase. This enzyme utilizes nicotinamide adenine dinucleotide phosphate (NADPH) during the glucose-to-sorbitol conversion, and it has been argued that cellular NADPH consumption by this pathway results in the accumulation of ROS [14]. This pathway leads to the depletion of NADPH, which is a required cofactor for the regeneration of reduced glutathione (GSH), an important scavenger of ROS [15]. Overexpression of human aldose reductase was demonstrated to be an accelerating factor of atherosclerosis in diabetic mice by reducing the expression of genes involved in glutathione regeneration [16]. The significance of the polyol pathway in hyperglycemic oxidative stress was also confirmed with diabetic rat lenses in several studies [17,18,19].

Third, diabetes has also been linked to the activation of the protein kinase C (PKC) pathway in producing ROS. An in vitro study involving aortic smooth muscle cells and endothelial cells demonstrated that chronic exposure of the cells to hyperglycemic conditions stimulated ROS production through PKC-dependent activation of NAD(P)H oxidase in both types of cells [20]. Another tissue that is vulnerable to this pathway is the retina, making it the main suspect as the cause of diabetic retinopathy [21]. The activation of PKC in the eye leads to macular edema, vascular leakage, capillary non-perfusion, and induction of growth factors such as the vascular endothelial growth factor (VEGF) [22]. VEGF causes proliferative diabetic retinopathy with macular edema, resulting in vision loss, retinal neovascularization, and excessive vascular permeability [23].

Fourth, increased non-enzymatic glycation occurs concurrently with hyperglycemia. In this pathway, high levels of intracellular glucose increase the glycation of amino groups of protein and binding reactive dicarbonyls, leading to AGE build-up [24]. Indeed, diabetes led to the accumulation of cardiac AGEs and caused cardiac abnormalities in streptozotocin (STZ)-induced diabetic rats [25]. Oxidative stress and glycation are closely related and are involved in glycoxidation, which generates ROS [15].

Fifth, hyperglycemia induces abnormal superoxide expression in the mitochondrial electron transport chain (mETC), which may cause an increase in ROS [26]. In the retinas of STZ-induced diabetic rats, elevated glucose concentration significantly increased mitochondrial superoxide due to the normalized superoxide production by an inhibitor of mETC complex II [27]. The overproduction of ROS by the mETC was suggested to be the key event resulting in the development of diabetic conditions [28].

Last, hyperglycemia induces impairment of the Nrf2/Kelch-like ECH-associated protein 1 (Keap1) pathway. The Nrf2/Keap1 pathway regulates various genes and functions involved in the regulation of oxidative stress, inflammation, recognition of DNA damage, and many more [29]. The Nrf2/Keap1 pathway is normally stimulated by oxidative stress and activated in a dose-dependent manner. Nrf2 is typically found in the cytosol, in association with its actin-bound inhibitor protein Keap1 under normal physiologic conditions. Upon activation by oxidative stress, newly synthesized Nrf2 translocates to the nucleus and heterodimerizes with a small musculoaponeurotic fibrosarcoma oncogene homolog (Maf) protein. The Nrf2/Maf complex activates antioxidant response elements (ARE) gene expression, which regulates several antioxidative and cytoprotective proteins. Nrf2 is otherwise inhibited by Keap1 in normal oxidative conditions [30]. The protein targets of Nrf2 include glutathione and its regeneration [31], superoxide dismutase [30], heme oxygenase-1 (HO-1) [32], and nuclear factor (NF-κB) [33]. Under hyperglycemic conditions, antioxidant enzymes regulated by Nrf2 had decreased expression in rat Müller retinal cells, including decreased catalase and glutathione activities [34]. Hyperglycemia seems to decrease the ability of Nrf2 to bind with the Maf protein located in the nucleus, overall leading to the Nrf2 target genes being downregulated [35].

The following sections of this review will address how the oxidative stress pathways are implicated in specific diabetes comorbidities. It will also highlight how treatments for diabetes that specifically target the Nrf2/Keap1 pathway have similar benefits on each comorbidity.

## 3. Diabetes and Cardiovascular Disease

### 3.1. Diabetes and Atherosclerosis

Though diabetic atherosclerosis is caused by various factors, the development of atherosclerotic lesions in diabetic patients is mainly associated with dyslipidemia due to an altered hepatic lipoprotein profile [36]. One of the most problematic types of modified lipoproteins with regards to atherosclerosis is small, dense low-density lipoprotein (sdLDL) because of its association with vascular plaque progression [37]. Being susceptible to oxidation due to the decreased content of antioxidative vitamins [38], sdLDL becomes a major accelerating factor of atherogenic inflammation. The glucose metabolism status is highly related to the plasma sdLDL level [39]. sdLDL is formed when excess triglyceride (TG) in LDL is hydrolyzed by hepatic lipase (HL), and a hyperglycemic environment makes the HL-mediated pathway more active [40]. Thus, glucose metabolism is highly related to the sdLDL content in blood plasma. In a study conducted in stable coronary artery disease patients with different glucose metabolism statuses (diabetes mellitus (DM), pre-DM, normal glycemia regulation), the results demonstrated that the plasma sdLDL level was significantly higher in diabetic and pre-diabetic participants compared to participants with normal glycemic regulation [39]. As sdLDL is a major factor in atherosclerosis development, this finding indicates that DM patients are more prone to atherosclerosis.

Another risk factor for atherosclerosis is impaired clearance of lipoproteins. A study conducted in diabetic and non-diabetic apolipoprotein-E-knockout (ApoE^−/−^) mice discovered that mice with an Ins2 gene mutation, and hence spontaneous type 1 diabetes (T1D), on an apoE-deficient background had decreased expression of lipolysis-stimulated lipoprotein receptor (LSR) in the liver [36]. LSR is responsible for remnant lipoprotein clearance through interaction with apoB [41,42]. In the mouse study, the T1D mice had increased atherosclerotic lesions in the aortic arch area and elevated plasma non-high-density-lipoprotein (non-HDL) cholesterol along with diminished LSR [36]. The connection between the histopathological and biochemical results suggests that diminished LSR expression exaggerates atherosclerosis and elevates non-HDL cholesterol in T1D mice by affecting apoB-mediated lipoprotein clearance. The study indicated defects in insulin receptor signaling in the liver as a possible major factor causing low LSR expression in T1D mice [36]. As mentioned previously, DM may become a major risk factor for atherosclerosis development via various mechanisms [36].

A substantial number of studies have demonstrated that DM medications such as metformin and liraglutide also have impacts on atherosclerosis. In the randomized REMOVAL (REducing with MetfOrmin Vascular Adverse Lesions) trial, DM patients treated with metformin for 3 years had a significant reduction of atherosclerosis, as measured by the progression of common carotid artery intima-media thickness (cIMT) at a follow-up visit [43]. The mechanisms by which metformin inhibits atherosclerosis can be categorized into three main types: activation of autophagy, disruption of angiotensin-II-type-1 receptor (AT1R), and upregulation of antioxidant enzyme superoxide dismutase 1 (SOD1). Metformin activates autophagy by activating AMPK, which is a key sensor for cellular ATP and a negative regulator of the mammalian target of rapamycin (mTOR) [44,45]. AMPK-mediated inhibition of mTOR induces autophagy in vascular endothelial cells and hence inhibits the progression of atherosclerosis [45,46]. In addition, research has suggested that metformin attenuates the expression of taurine up-regulated gene 1 (TUG1), a long non-coding RNA (lncRNA) known to repress autophagy and promote atherosclerosis progression, in human umbilical vein endothelial cells [45,47,48]. A study conducted on mice fed a high-fat diet (HFD) showed that metformin-treated mice had a significantly lower expression of AT1R [49]. Attenuating the expression of AT1R inhibits the binding of angiotensin II, which is known to play a major role in the senescence of vascular smooth muscle [49,50]. In the HFD mouse study, metformin-treated mice indeed had significantly smaller plaque areas and lower vascular senescence levels in the aortas [49]. Furthermore, the study also demonstrated that metformin-treated mice on a regular chow diet had significantly higher aortic expression levels of SOD1 in comparison to non-metformin-treated mice on the same diet ([49], Table 1). Upregulation of SOD1 results in the inhibition of reactive oxidative species (ROS), which causes oxidative stress [51].

Liraglutide, a glucagon-like peptide-1 (GLP-1) receptor agonist, significantly attenuated atherogenesis and improved plaque stability in ApoE^−/−^ mice [65]. Similarly, liraglutide reduced aortic atherosclerotic plaque lesion size in LDL receptor-deficient (LDLr^−/−^) mice [66]. Furthermore, the cMIT of patients with impaired glucose tolerance (IGT) treated with liraglutide was significantly smaller than that of patients treated with lifestyle intervention [67]. There are several suggested mechanisms by which liraglutide attenuates atherosclerosis. In a study utilizing human monocyte-derived macrophages, liraglutide demonstrated a significant suppressive effect on foam cell formation via the downregulation of acyl-coenzyme A:cholesterol acyltransferase 1 (ACAT1), an important factor in early atherogenesis. Liraglutide was also shown to suppress foam cell formation by reducing the uptake of oxidized LDL (oxLDL) and cholesterol in ApoE^−/−^ mice [53]. Another in vivo study explained the anti-atherosclerotic effects of incretin-based drugs like liraglutide through their ability to downregulate CD36 in macrophages [54].

*Cinnamomum verum* J. Presl., Lauraceae (*C. verum*), commonly known as cinnamon, is known not only for its versatility as a spice but also for its anti-diabetic effects, reducing glycemic and lipidemic levels in patients with T2D [68,69]. Similar to metformin and liraglutide, *C. verum* has also demonstrated notable anti-atherosclerotic properties in various studies. The aqueous extract of *C. verum* effectively prevented the thickening of aorta and reduced the risk of atherogenicity in dexamethasone-induced atherosclerotic rats, as evidenced by increased HDL and decreased TG and LDL [70]. The aqueous extract of *Cinnamomum cassia* (L.) J. Presl., Lauraceae (*C. cassia*), another species that is also widely referred to as cinnamon, suppressed atherosclerosis development by preventing apoA-1 glycation and downregulating cholesteryl ester transfer protein (CETP) in hypolipidemic zebrafish both in vivo and in vitro [71]. In another study, the aqueous extract of *C. cassia* successfully demonstrated its anti-inflammatory properties by inhibiting the expressions of CD36 and scavenger receptor class A (SRA), while regulating extracellular signal-related kinase (ERK) 1/2 activity and monocyte differentiation in macrophages, confirming its potential as an effective anti-atherosclerotic substance [57].

### 3.2. Nrf2-Targeting Treatment in Diabetic Atherosclerosis

As explored in the abovementioned studies, oxidative stress is closely related to atherosclerosis progression. A considerable number of studies have investigated the correlation between atherogenesis and the Nrf2/Keap1 pathway. Such correlation has been corroborated by transplanting Nrf2-knockout (Nrf2^−/−^) bone marrow to LDLr^−/−^ mice. Macrophages isolated from both wild-type (WT) and Nrf2^−/−^ bone marrow groups were incubated with native LDL and different modified LDLs to examine the LDL uptake and gene expression of macrophages. The results showed that the loss of Nrf2 in macrophages increased the uptake of modified LDL, the expression of scavenger receptors (CD36, SRA, LOX-1, CXCL16, TLR4), and the expressions of pro-atherogenic inflammatory cytokines (MCP-1, TNFα, and IL-6). Further, the cross-sectional lesion areas in the aortic roots of the mice with Nrf2^−/−^ bone marrow were significantly larger than those of WT bone marrow mice [72]. ROS causes oxidative stress in arterial intima by oxidizing LDL in the arterial wall. The oxidized LDL becomes a ligand for scavenger receptors, causing dysregulated uptake of modified LDL into intimal macrophages. These macrophages then convert into foam cells containing substantial amounts of cholesterol esters and contribute to both early and late atherosclerotic lesions ([73], Figure 1). Thus, Nrf2 plays a major role in atherosclerotic inflammation due to oxidative stress.

In addition to the upregulation of SOD1 expression, metformin also attenuates oxidative stress by enhancing the Nrf2/Keap1 signaling pathway. In a study conducted on an HFD mouse model, significant upregulations in Nrf2 and HO-1 expression and downregulations in Keap1 expression were observed in the cardiac tissue of the HFD mice treated with metformin. In addition, the study also suggested that metformin activated Nrf2 translocation into the nucleus in the cardiac tissue. Such findings seemed to have a notable connection with the protective characteristic of metformin against cardiac tissue remodeling due to HFD-induced obesity, which was also observed in the study [52]. Therefore, metformin enhances endogenous antioxidant activities by promoting the Nrf2/Keap1 pathway. Since the HFD mouse study focuses on the antioxidant effect of metformin on cardiac tissue, further research on the Nrf2/Keap1-associated antioxidant effect of metformin on vascular endothelium is required for the investigation of its association with atherosclerosis.

The anti-atherosclerotic and antioxidative properties of liraglutide are also related to the Nrf2 pathway. In STZ-induced diabetic rats, liraglutide showed its protective effects on brain nerve cells against cerebral ischemia by activating the Nrf2/HO-1 pathway and promoting the activities of other antioxidant enzymes, including SOD and myeloperoxidase (MPO) [55]. Liraglutide also improved the angiogenic capability of the endothelial progenitor cell (EPC) and ameliorated ischemic angiogenesis in diabetic mice by increasing the activity of Nrf2 [56]. While there is limited information about liraglutide’s impact on Keap1, these findings suggest liraglutide’s potential as an effective Nrf2-related antioxidative treatment for diabetes-induced atherosclerosis.

Cinnamaldehyde (CN), a main bioactive component of *C. verum* and *C. cassia*, becomes an effective antioxidant by enhancing Nrf2 nuclear translocation [58]. In HFD-induced metabolically dysfunctional rats, CN induced a significant upregulation of Nrf2 at the vascular level, which then led to the amelioration of vascular oxidative damage and endothelial dysfunction [59]. CN also played a role as a powerful activator of the Nrf2-dependent antioxidative response while maintaining the Keap1 levels in human epithelial colon cells [60]. Taken together, CN in C. *verum* and *C. cassia* has been confirmed to be a notable anti-diabetic and anti-atherosclerotic substance that can target the Nrf2/Keap1 signaling pathway.

### 3.3. Diabetes and Cardiomyopathy

Diastolic dysfunction is one of the most prominent cardiac complications commonly found in patients with diabetes [74]. It frequently follows myocardial infarction (MI), which is another common cardiac complication causing mortality in patients with diabetes [75,76]. One suggested mechanism through which hyperglycemic conditions cause diastolic dysfunction is by altering nuclear O-linked-N-acetylglucosaminylation (O-GlcNAcylation) in myocytes [77]. O-GlcNAcylation is characterized by posttranslational modification, which results in the addition of O-linked-β-N-acetylglucosamine (O-GlcNAc) to serine and threonine residues of proteins [78]. Protein O-GlcNAcylation, associated with increased glucose flux through the hexosamine biosynthesis pathway, is known to be involved in various pathological disorders, such as altered insulin resistance and inflammation [79].

In an in vitro study using neonatal rat cardiomyocytes to investigate the association between O-GlcNAcylation and diastolic dysfunction, cardiomyocytes cultured in high glucose (25 mM) resulted in an extended duration of calcium (Ca^2+^) transients compared to myocytes cultured in normal glucose (5.5 mM). In addition, myocytes exposed to high glucose had reduced sarcoendoplasmic reticulum Ca^2+^-ATPase 2a (SERCA2a) mRNA and protein expression, which is responsible for Ca^2+^ sequestration and the induction of diastolic relaxation. SERCA2a is a major sarcoplasmic reticulum (SR) enzyme responsible for Ca^2+^ uptake into the SR and is regulated by endogenous protein phospholamban (PLN) [80]. High-glucose-treated myocytes also exhibited diminished SERCA2a promoter activity and increased O-GlcNAcylation of nuclear proteins. The study also discovered the connection between O-GlcNAcylation and SERCA2a by treating high-glucose-exposed myocytes with adenovirus encoding O-GlcNAcase (GCA), an enzyme responsible for removal of O-linked N-acetylglucosamine residues [81], or adenovirus encoding O-GlcNAc-transferase (OGT), an enzyme responsible for the O-linkage of a single acetylglucosamine to serine and threonine [82]. As a result, high-glucose myocytes treated with GCA had improved myocytic Ca^2+^ transients and SERCA2a protein levels, whereas high-glucose myocytes treated with OGT had prolonged Ca^2+^ transient decays and decreased SERCA2a protein levels. With or without OGT adenovirus, high-glucose myocytes demonstrated increased levels of O-GlcNAcylated specificity protein 1 [77]. Such results indicate that O-GlcNAcylation plays an important role in the impairment of Ca^2+^ cycling in cardiomyocytes, hence affecting diastolic function. Furthermore, the connection between O-GlcNAcylation and the Nrf2/Keap1 signaling pathway, discussed in the following subsection, adds to the significance of this mechanism.

Empagliflozin, a sodium-glucose cotransporter-2 (SGLT-2) inhibitor, has been determined to be effective for diastolic dysfunction via experiments involving both human and non-human tissue. Previous research has suggested the significant impact empagliflozin has on left ventricular diastolic function by demonstrating a difference in the levels of phosphorylated PLN and the SERCA2a/PLN ratio. The study examined left ventricular tissue obtained from type 2 DM (T2DM) obese mice treated with either a vehicle-containing diet or an empagliflozin-containing diet. Although there was no significant change in SERCA2a expression levels between the two groups, the phosphorylation of PLN and the SERCA2a/PLN ratio were significantly increased in the presence of empagliflozin. As phosphorylated PLN increases SERCA2a activity [80], the results demonstrate the enhancing property of empagliflozin on SR Ca^2+^ cycling, and thus cardiac diastolic performance [83]. The effect of empagliflozin on diastolic dysfunction has also been corroborated by a study conducted with human failing myocardium. After the wash-in of empagliflozin, trabeculae from human hearts in end-stage heart-failure had a significant reduction in abnormally high diastolic tension. In addition, the administration of empagliflozin significantly attenuated myofilament stiffness of skinned cardiomyocytes from patients with heart failure with preserved ejection fraction (HFpEF). This effect was achieved by increasing the phosphorylation of myofilament regulatory proteins [84]. Such findings are consistent with the previously mentioned empagliflozin study on mouse myocardium.

Sitagliptin, a dipeptidyl peptidase-4 (DPP-4) inhibitor, has also been suggested as an effective anti-diabetic medication for cardiomyopathy. In patients with IGT, acarbose was associated with major reductions in the risk of MI and other cardiovascular events [85]. In Goto-Kakizaki rats with T2D, treatment with sitagliptin attenuated cardiac fibrosis and hypertrophy in the left ventricular (LV) myocardium [86]. In addition, sitagliptin inhibited oxidative stress in H9c2 cardiomyocytes [87] and downregulated expressions of myocardial pro-inflammatory cytokines in STZ-induced diabetic mice [62]. These findings confirm the potential of sitagliptin not only as an anti-cardiomyopathy treatment in patients with diabetes but also as an effective antioxidant that can resolve many more conditions beyond diabetic cardiomyopathy.

### 3.4. Nrf2-Targeting Treatment in Diabetic Cardiomyopathy

Excessive oxidative stress becomes a major inducer of diabetic cardiomyopathy [88]. Diabetes-induced oxidative stress, in particular, has been known to have significant impacts on both the diastolic and systolic capabilities of the heart. Previous research has discovered that diabetes aggravates oxidative stress and weakens antioxidant defense in the heart. The results demonstrated a significant increase in cardiac ROS and fibrosis, along with decreases in Nrf2 and HO-1 expressions, in the cardiac tissue of rats with T1D and/or myocardial ischemia/reperfusion (IR). In addition, the T1D+IR rat cardiac tissue exhibited a higher concentration of lipid peroxidation product malondialdehyde and a larger infarct size compared to IR tissue. Such changes were more prominent in tissue from rats with both T1D and IR compared to rats with just T1D or IR. In addition, the results of the cardiac hemodynamic examination exhibited increased left ventricular end-diastolic pressure and decreased left ventricular systolic pressure in T1D rats, indicating impaired left ventricular contractile function. Thus, these experimental results suggest that diabetes-induced oxidative stress causes impairment in cardiac function, as marked by diastolic and systolic dysfunction [89].

Interestingly, the aforementioned O-GlcNAcylation under hyperglycemic conditions is related to the Nrf2/Keap1 signaling pathway. Biochemical connections between O-GlcNAcylation and the Nrf2/Keap1 signaling pathway have been studied, confirming that α-lipoic acid decreases global O-GlcNAcylation and increases Nrf2 nuclear translocation in STZ-induced diabetic rats [90,91]. More recent studies have discovered that inhibition or genetic knockdown of O-GlcNAc transferase (OGT) induces activation of the Nrf2 pathway. Furthermore, it has been found that Keap1 interacts with OGT and is O-GlcNaclyated. This process leads to reduced Nrf2 ubiquitination and inhibition of the Nrf2 pathway [92,93]. Thus, O-GlcNAcylation is crucial for explaining the impact of the Nrf2/Keap1 pathway on diabetic cardiomyopathy.

The protective property of empagliflozin against cardiomyopathy, including diastolic dysfunction, can be explained by its association with Nrf2 as well. A recent finding has demonstrated that empagliflozin attenuates oxidative stress and improves mitochondrial function in myocardial tissue via activation of the Nrf2 signaling pathway. A study found that the expression levels of Nrf2 and SOD2 were significantly increased in the cardiac tissue of empagliflozin-treated diabetic mice, in comparison to that of vehicle-treated diabetic mice. Furthermore, empagliflozin was also determined to be a promoting factor of Nrf translocation into the nucleus of cardiomyocytes. Accordingly, the heart tissues of empagliflozin-treated diabetic mice showed significantly reduced intracellular and mitochondrial ROS levels. In turn, empagliflozin successfully inhibited diabetes-induced fibrosis of myocardial tissue [61]. As cardiac fibrosis is an inducing factor of diastolic dysfunction [94], empagliflozin can be suggested as an Nrf2/Keap1-associated antioxidant treatment for diastolic dysfunction.

The antioxidative mechanism of sitagliptin is also connected with the Nrf2/Keap1 pathway. In mice with acute pancreatitis, sitagliptin worked as an effective antioxidative and ameliorated pancreatitis-induced lung injury by activating the p62-Nrf2-Keap1 signaling pathway, promoting Nrf2 nuclear translocation. In addition, the study confirms the direct relationship between sitagliptin and NRf2, demonstrating the reduced anti-inflammatory effect of sitagliptin in Nrf2-knockout mice [63]. Sitagliptin was also found to upregulate the levels of Nrf2 and its downstream protein HO-1 in the small intestine tissues of mice with radiation-induced intestinal damage [64]. Thus, sitagliptin is worthy of close attention not only as an anti-diabetic medication but also as a potent treatment for diabetic cardiomyopathy that targets the Nrf2/Keap1 signaling pathway.

## 4. Diabetes and Kidney Disease

Diabetes can cause renal impairment, leading to diabetic nephropathy (DN). DN manifests as a decline in kidney function associated with persistent microalbuminuria [95]. Elevated albumin excretion has been shown to be predictive of subsequent DN in patients with T1D [96]. In the development of DN, diabetic patients have also been shown to have higher glomerular filtration rates and blood pressure levels [96]. Sustained hyperglycemia in diabetes enhances the production of ROS through activation of the polyol pathway, hexosamine pathway, PKC, and advanced glycation [15]. These processes together exacerbate DN. Untreated DN leads to the progression of chronic kidney disease and, subsequently, end-stage kidney disease [97].

Sodium-glucose transport protein 2 (SGLT-2) inhibitors, including dapagliflozin, canagliflozin, and empagliflozin, are synthetic drugs that are used to reduce hyperglycemia and have been shown to have protective effects against DN. A study on empagliflozin demonstrated its effect in lowering glucose levels in diabetic mice while also preventing glomerular hyperfiltration [98]. Empagliflozin can also decrease albuminuria levels, as well as renal growth and inflammation [98]. However, SGLT-2 inhibitors can also have adverse effects, such as hypotension and genital mycotic infections [99]. A recent retrospective cohort study, including 2083 kidney transplant recipients with diabetes and more than six months post-transplant, demonstrated a 48% decrease in the risk of developing composite outcomes such as all-cause mortality, death-censored graft failure, and serum creatinine doubling in patients taking SGLT2 inhibitors compared to those not taking them. The mean follow-up duration was 62.9 ± 42.2 months. The majority of patients with the SGLT2 inhibitor had a stable estimated glomerular filtration rate (eGFR), without the initial drop, and only 15.6% of the SGLT2 inhibitor group showed an acute drop in eGFR during the first month, but the eGFR recovered thereafter [100].

Although there are few synthetic drugs for the treatment of DN, *Aloe vera* (L.) Burm.f., Asphodelaceae (*A. vera*) (300 mg/day orally) was shown to be effective in reducing oxidative stress and lipid profile in rats with diabetes-induced nephropathy ([101], Table 2). This treatment led to lower levels of urea nitrogen and creatinine in the blood. *A. vera* also diminishes urinary albumin excretion and ameliorates glomerular hypertrophy and mesangial matrix expansion [101]. Other studies have also shown that the leaf gel of *A. vera* possesses antioxidant properties [102]. Through renal protective action, *A. vera* prevents the destruction of the kidney’s morphology [102].

*Ginkgo extractum siccum raffinatum et quantificatum*, commonly known as ginkgo biloba extract (GBE), can also prevent DN by activating the Nrf2/HO-1 pathway. At the same time, Gbe has been proven to carry out antihyperglycemic activities in STZ-induced chronic diabetic rats [104]. In a study, STZ-induced diabetic mice that eventually developed DN were treated with Gbe to examine its renal protective effects [103]. This study found that GBE enhanced HO-1 and Nrf2 expression [103]. However, when Nrf2 siRNA was administered, it suppressed the activation of HO-1 and Nrf2 [103]. This confirmed that GBE works through Nrf2-mediated HO-1 activation and has the potential application for treating DN [103].

Metformin in combination with L-ergothioneine (L-egt) has been shown to improve the therapeutic outcome of DN. A study tested this combination and its effect on renal damage using a T2D rat model. Four groups of diabetic rats were divided and treated as follows: non-diabetic control (NC)-1 mL/100 g body weight (bwt) of distilled water, diabetes control (DC)-1 mL/100 g bwt of distilled water, diabetes + L-egt (DE)-35 mg/kg bwt of L-egt, diabetes + metformin (DM)-500 mg/kg bwt of metformin, and diabetes + L-egt + metformin (DEM). Ultimately, L-egt increased Nrf2 protein expression with or without metformin [105]. The NC and DC groups, however, showed only low levels of Nrf2 protein expression. The presence of Nrf2 enhances the antioxidant defense system by upregulating cytoprotective genes and antioxidant enzymes and downregulating the TGF-β1 gene. Nrf2 also reduces inflammatory cytokines through inhibition of the nuclear transcription factor-κB (NF-kB) inflammatory gene. Thus, the combination of metformin and L-egt does not only reduce glucose levels, but also alleviates renal damage by promoting better antioxidant defense and reducing renal inflammation [105].

## 5. Diabetes and Liver Diseases

### 5.1. Diabetes and Progression of Non-Alcoholic Liver Disease

T2D and nonalcoholic fatty liver disease (NAFLD) are common co-existing diseases and can synergize in causing adverse outcomes [106]. NAFLD is characterized by fat accumulation in the liver. Unresolved NAFLD can lead to nonalcoholic steatohepatitis (NASH), which can further progress to liver cirrhosis and hepatocellular carcinoma [107]. A study assessing biopsies for the progression of NAFLD showed that as the metabolic severity of diabetes progresses, significant NAFLD progression also occurs [108]. While the presence and progression of both NAFLD and T2D are common findings, it remains unclear if one disease leads to the other. There is strong evidence that NAFLD significantly increases the risk of diabetes by up to 5-fold [106]. However, the reverse has not been conclusively proven. T2D patients are usually diagnosed with NAFLD significantly long after the onset of their diabetes. In addition, liver fat is increased in patients with T2D compared to healthy controls [109]. A different study in non-diabetic subjects, however, showed that increased HbA1c and insulin resistance are correlated with an increased risk of NAFLD, independent of obesity [110].

Interestingly, several T2D medications have been associated with improved NAFLD outcomes [111]. Metformin is a first-line agent used to improve insulin sensitivity in diabetic patients [106]. In addition to improving insulin sensitivity in the liver, metformin also reduces body fat, increasing fatty acid oxidation and reducing lipogenesis [112]. However, metformin has not been approved for off-label use in NAFLD patients without diabetes at this time [106].

Thiazolidinediones are a class of medications that improve insulin sensitivity in the adipose tissue by activating the proliferator-activated receptor γ (PPARγ) pathway. In a clinical trial of pioglitazone involving patients with NAFLD and either T2D or pre-DM, this drug improved the fibrosis score, reduced triglyceride levels in the liver, and improved adipose tissue, liver, and muscle insulin sensitivity when compared to a placebo [113].

GLP-1 analogs, such as exenatide, directly inhibit lipogenesis in the liver and improve insulin sensitivity through the proliferator-activated receptor δ (PPARδ) pathway [114]. Liraglutide has also been shown to improve NASH outcomes in T2DM patients after one year, compared to a placebo [111].

Newer methods addressing liver cirrhosis have focused on the dysfunction of the Nrf2/Keap1 pathway. One study showed that endogenous opioid levels increase with liver cirrhosis, a condition also associated with a significant increase in thrombospondin-1 (THBS-1) and Keap1, as well as a reduction in Nrf2 ([115], Table 3). As such, it has been suggested that the opioid antagonist naltrexone may have protective effects on the liver. In a bile duct ligation (BDL) liver cirrhosis rat model, naltrexone significantly reduced hepatic necrosis and lobular damage. BDL rats treated with naltrexone showed a significant reduction in THBS-1 and NADPH oxidase-1 (NOX-1), while also exhibiting an increase in Nrf2 levels. NOX-1 is a primary producer of ROS that has been shown to be activated by THBS-1 [116]. Control BDL rats showed significantly increased levels of THBS-1 and NOX-1 and reduced levels of Nrf2 [117].

The results above are also supported by a study using Nrf2 knockout mice, which showed that these mice are vulnerable to more severe NASH and cirrhosis when compared to wild-type mice. In both groups, they were fed HFD for 24 weeks. Nrf2 knockout resulted in greater induction of lipogenic genes, lower expression of β-oxidation genes, and diminished acetyl coenzyme A (CoA) activity, resulting in greater fatty acid synthesis. Antioxidant response, normally present in wildtype liver, was missing in the Nrf2 knockout mice [129].

Surprisingly, drugs for T2D have also been shown to affect the Nrf2 pathway in the liver. In addition to its effect on liver insulin sensitivity, metformin increases Nrf2 and HO-1 expression in the liver [118]. Liraglutide similarly has this effect, and using an obese mouse model, it was found that liraglutide acts through the Sestrin2-mediated signaling to increase Nrf2 and HO-1 expression in the liver. In this pathway, liraglutide caused an increase in Sestrin2 expression, which then increased the autophagic degradation of Keap1 and increased the expression of Nrf2 [119].

Empagliflozin, an SGLT-2 inhibitor used to improve glycemic control in T2D, has also demonstrated a protective effect in the liver. A study on pre-diabetic non-obese hereditary hypertriglyceridemic (HHTg) rats, when compared to Wistar controls, exhibited significantly increased triglyceride levels, IGT, and increased levels of pro-inflammatory cytokines, including leptin, MCP-1, TNFα, and IL-6. HHTg rats also accumulated ectopic hepatic lipids and toxic intermediates, leading to increased oxidative stress and reduced antioxidant enzyme levels. Empagliflozin treatment reduced the accumulation of neutral and toxic acylglycerols in the HHTg rat liver, accompanied by significant increases in Nrf2 and normalization of cytochrome p450 expression. This treatment resulted in improvement in liver lipid metabolism [120].

### 5.2. Diabetes and Alcoholic Liver Disease

Alcohol consumption has been associated with an increased risk of T2D, especially when combined with pre-existing fatty liver disease (FLD). Higher diabetic risk is also seen among moderate alcohol consumers without FLD [130]. The severity of liver disease is significantly increased with alcohol consumption in synergy with insulin resistance and total-to-LDL cholesterol ratio [131].

Alcohol disrupts normal cholesterol metabolism in the liver, causing esterification and producing triglycerides, phospholipids, and cholesterol esters [132]. Alcoholic liver disease (ALD) has been shown in mice models to be associated with the downregulation of Nrf2, while upregulating Keap1, a negative regulator of Nrf2 function. *Dracocephalum tanguticum* Maxim., Lamiaceae (*D. tanguticum*) is a traditional Chinese medicine previously used in patients with ALD and hepatitis. A test of its ethyl acetate extract (DtM-E), resulted in increased Nrf2 activation and downregulated Keap1 expression in the ALD mice model [121]. This study suggests that DtM-E has protective effects on ALD through the inhibition of the oxidative stress and inflammatory injury pathways, which have been previously shown to be key factors leading to the progression of hepatic inflammation, fibrosis, and liver cancer [133,134].

Dihydromyricetin (DMY) has also been shown to be effective in the ALD mice model, as it attenuates liver lipid peroxidation, triglyceride deposition, and inflammatory cytokine elevation. Treating ALD mice with DMY had hepatoprotective effects by reducing the expression of Keap1 and HO-1, alleviating disordered localization of NF-κB and Nrf2, and increasing Nrf2 expression [122]. As a flavonoid, DMY now joins resveratrol [135] and quercetin [136], which have previously been shown to activate the Nrf2 pathway, leading to the suppression of liver lipid synthesis and inflammation. Resveratrol was previously shown to have Nrf2 pathway-mediated benefits in diabetes by reducing blood glucose, HbA1c, and insulin resistance [137]. Treatment with the Nrf2 inhibitor, retinoic acid, prevents resveratrol from exerting these beneficial effects.

More recently, exercise has shown protective effects on the liver in the face of ethanol-induced damage. Regular exercise has been previously shown to produce antioxidative and anti-inflammatory effects, especially in diabetic patients [138]. This recent study showed that these protective effects were mediated by increased activity of the Nrf2 and HO-1 pathways, and decreased Keap1 expression in the liver, kidney, and heart tissues [139]. In addition, Nrf2 and HO-1 expression have been positively linked to levels of HDL and atherogenic index [140]. The increased Nrf2 and HO-1 expression, as well as the inhibition of Keap1, led to improved levels of liver oxidative stress [139].

### 5.3. Effects of Diabetes Drugs on Liver Responsse to Toxins

The liver is the primary organ responsible for detoxification of the body by converting toxic molecules into less toxic metabolites, which can then be converted into water-soluble metabolites discarded in urine [141]. The detoxification process also generates antioxidants, such as glutathione (GSH), which protect the liver itself from oxidative stress during the process [141].

Detoxification of certain toxins can result in reduced Nrf2 expression, while increasing Keap1 and NF-κB ([123], Figure 2). Toxins that have been shown to have these effects include methotrexate (a chemotherapy medication) [123], paraquat (a pesticide) [124], and carbon tetrachloride (CCl_4_; a chemical used in the production of refrigerants) [125,126].

Toxin-induced alterations of protein expression resulted in decreased GSH content and SOD activity, as well as increased malondialdehyde (MDA) and nitrite (NO_2_^−^) content [123]. GSH and SOD are the primary antioxidant molecules and enzymes, respectively, responsible for resolving oxidative stress. These changes resulted in the increased oxidative stress burden on liver cells, leading to apoptosis [123] and fibrosis [126].

Several drugs have been shown to counter these negative effects, including diallyl disulfide [123], ellagic acid [124], ursolic acid [125], monoammonium glycyrrhizinate and cysteine hydrochloride [126]. These drugs work by upregulating Nrf2 and reducing Keap1 production, significantly ameliorating toxicity in the liver. There may be a role for the use of these drugs as adjuncts to therapeutic regimens containing hepatotoxic drugs, especially in the treatment for cancer, in the future [123].

There are several inhibitors of the Nrf2/Keap1 signaling pathway. Luteolin (3′,4′,5,7-tetrahydroxyflavone) is a polyphenolic flavonoid that has been found to be an effective inhibitor of the Nrf2 pathway [142,143,144]. Other flavonoids, such as apigenin (4′,5,7-trihydroxyflavone) [145], wogonin (5,7-dihydroxy-8-methoxyflavone) [146], and chrysin (5,7-dihyxroxyflavone) [147] have also been confirmed as potent Nrf2 inhibitors. As anti-proliferative and anticancer agents, these compounds are reported to be effective in reversing drug resistance by inhibiting the Nrf2 signaling pathway [148]. Interestingly, however, luteolin used with metformin activates the Nrf2 pathway in the liver and protects the organ against toxic metabolite damage. Metformin, by itself or in combination with luteolin, upregulates Nrf2 and HO-1 expression and rescues the liver from CCl_4_ toxicity [118,127]. Liraglutide, in addition to the Sestrin2-mediated Nrf2 pathway activation previously mentioned [119], also increased the transcription of downstream phosphorylated cAMP response element-binding protein (pCREB), in a methotrexate toxicity rat model [128]. Methotrexate treatment reduces both pCREB and Nrf2 levels. pCREB activation increased the acetylation of Nrf2, increasing its sequence-specific DNA binding capacity, thus increasing transcription of its downstream genes [128]. This protective effect was also seen in CCl_4_ toxicity in mice, with liraglutide treatment reducing acute liver injury [149].

## 6. Conclusions

The Nrf2/Keap1 signaling pathway is a profoundly studied defense mechanism against oxidative stress and inflammation. With the significance of oxidative stress in diabetes confirmed, the protective properties of the Nrf2/Keap1 activation become an important strategy for the resolution of hyperglycemia-induced oxidative stress and organ damage.

This review demonstrated that the Nrf2/Keap1 signaling pathway is a common mechanism implicated in the pancreas of diabetic patients, and similarly in the organs impaired in common diabetic comorbidities, including the heart, kidney, and liver. Furthermore, the potential of common anti-diabetic medications and substances as activating agents of the Nrf2/Keap1 signaling pathway was investigated in detail. Finally, by bridging these two realms, this review aims to suggest the potential of medications and substances that can be used not only as anti-diabetic treatments but also as multi-organ treatments that effectively target the Nrf2/Keap1 pathway and resolve diabetic comorbidities.

As a potent antioxidant target that can be a key to the treatment of diabetes and various diabetic comorbidities, the Nrf2/Keap1 signaling pathway remains worthy of further investigation. This raises an opportunity for clinicians to emphasize the importance of blood glucose control in preventing the spread of Nrf2/Keap1 pathway-induced impairment to other organs and the consequent onset or progression of diabetic comorbidities. In addition, this pathway presents additional opportunities for the development of treatments targeting multi-organ diseases associated with diabetes.

## Figures and Tables

**Figure 1 ijms-25-00821-f001:**
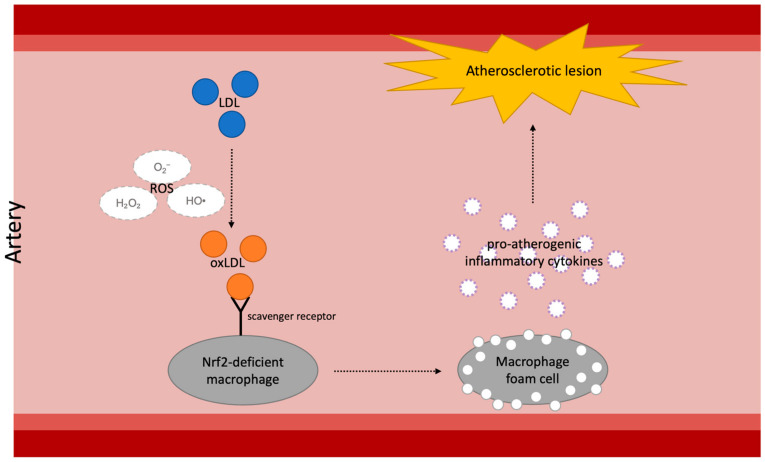
The overall mechanism of the development of atherosclerosis caused by Nrf2 deficiency in arterial macrophages: LDL, low-density lipoprotein; ROS, reactive oxygen species; oxLDL, oxidized low-density lipoprotein; Nrf2, nuclear factor erythroid 2–related factor 2.

**Figure 2 ijms-25-00821-f002:**
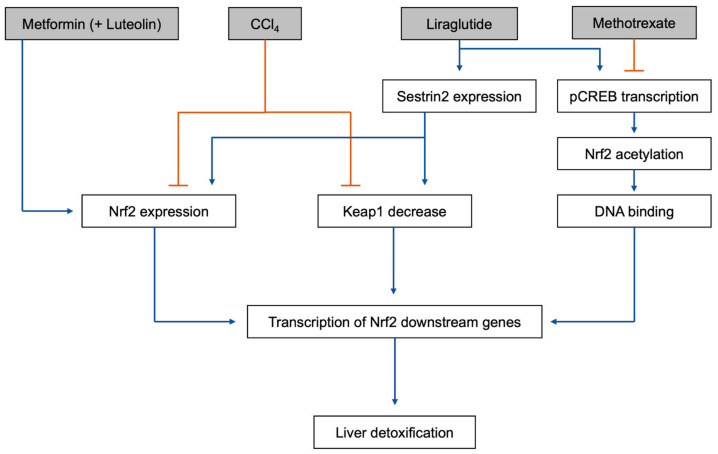
The mechanisms of metformin (by itself or with luteolin) and liraglutide detoxifying the liver through the Nrf2/Keap1 signaling pathway, as well as the mechanisms of CCl4 and methotrexate inducing liver toxication by inhibiting the Nrf2/Keap1 signaling pathway: pCREB, phosphorylated cAMP response element-binding protein; Nrf2, nuclear factor erythroid 2–related factor 2; Keap1, Kelch-like ECH-associated protein 1.

**Table 1 ijms-25-00821-t001:** Table of drugs/substances that can be used for cardiovascular diabetes comorbidities and their antioxidative actions: SOD1, superoxide dismutase 1; ROS, reactive oxidative species; Nrf2, nuclear factor erythroid 2–related factor 2; Keap1, Kelch-like ECH-associated protein 1; HO-1, heme oxygenase-1; oxLDL, oxidated low-density lipoprotein; SOD, superoxide dismutase; MPO, myeloperoxidase; *C. verum*, *Cinnamomum verum* J. Presl., Lauraceae; *C. cassia*, *Cinnamomum cassia* (L.) J. Presl., Lauraceae (*C. cassia*); SRA, scavenger receptor class A; SOD2, superoxide dismutase 2.

Disease	Drug/Substance	Action in Nrf2 and Other Antioxidative Pathways	References
Atherosclerosis	Metformin	SOD1 upregulation	[49,51]
		ROS inhibition	
		Nrf2 upregulation	[52]
		Keap1 downregulation	
		HO-1 upregulation	
		Upregulation of Nrf2 translocation	
	Liraglutide	Inhibition of oxLDL uptake	[53]
		CD36 downregulation	[54]
		Nrf2/HO-1 pathway activation	[55]
		Promotion of SOD and MPO activities	
		Promotion of Nrf2 activities	[56]
	*C. verum/C. cassia*	CD36 downregulation	[57]
		SRA downregulation	
		Upregulation of Nrf2 translocation	[58]
		Nrf2 upregulation	[59]
		Keap1 regulation	[60]
Cardiac fibrosis	Empagliflozin	Nrf2 upregulation	[61]
		SOD2 upregulation	
		Upregulation of Nrf2 translocation	
	Sitagliptin	Downregulation of inflammatory cytokines	[62]
		Upregulation of Nrf2 translocation	[63]
		Nrf2 upregulation	[64]
		HO-1 upregulation	

**Table 2 ijms-25-00821-t002:** Table of drugs/substances that can be used for kidney diabetic comorbidities (e.g., diabetic nephropathy) and their antioxidative actions: *A. vera*, *Aloe vera* (L.) Burm.f., Asphodelaceae; GBE, *Ginkgo extractum siccum raffinatum et quantificatum*, commonly known as ginkgo biloba extract; L-egt, L-ergothioneine; TGF-β1, transforming growth factor beta-1; NF-kB, nuclear transcription factor-κB.

Disease	Drug/Substance	Action in Nrf2 and Other Antioxidative Pathways	References
Diabetic nephropathy	*A. vera*	Reduction of oxidative stress	[102]
	GBE	Nrf2 upregulation	[103,104]
		HO-1 upregulation	
	Metformin with L-egt	Nrf2 upregulation	[105]
		Upregulation of cytoprotective genes and antioxidative enzymes	
		Downregulation of TGF-β1 gene	
		Inhibition of the NF-kB inflammatory gene	

**Table 3 ijms-25-00821-t003:** Table of drugs/substances that can be used for liver diabetic comorbidities and their antioxidative actions: *D. tanguticum*, *Dracocephalum tanguticum* Maxim., Lamiaceae; THBS-1, thrombospondin-1; NOX-1, nicotinamide adenine dinucleotide phosphate oxidase-1; pCREB, phosphorylated cAMP response element-binding protein.

Disease	Drug/Substance	Action in Nrf2 and Other Antioxidative Pathways	References
Nonalcoholic fatty liver disease	Naltrexone	Nrf2 upregulation	[115]
		THBS-1 downregulation	
		NOX-1 downregulation	
	Metformin	Nrf2 upregulation	[118]
		HO-1 upregulation	
	Liraglutide	Nrf2 upregulation	[119]
		HO-1 upregulation	
		Sestrin2 upregulation	
		Keap1 degradation	
	Empagliflozin	Nrf2 upregulation	[120]
		Normalization of cytochrome p450 expression	
Alcoholic liver disease	*D. tanguticum*	Nrf2 upregulation	[121]
		Keap1 downregulation	
	Dihydromyricetin	Nrf2 upregulation	[122]
		Keap1 downregulation	
		HO-1 downregulation	
		Alleviation of disordered NF-κB and Nrf2 localization	
Toxin-induced liver damage	Diallyl disulfide	Nrf2 upregulation and Keap1 downregulation	[123]
	Ellagic acid		[124]
	Ursolic acid		[125]
	Monoammonium glycyrrhizinate		[126]
	Cysteine hydrochloride		
	Metformin with luteolin	Nrf2 upregulation	[118,127]
		HO-1 upregulation	
	Liraglutide	pCREB upregulation	[128]

## Data Availability

Not applicable.

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
