# Peer review of "Nrf2 Pathway and Oxidative Stress as a Common Target for Treatment of Diabetes and Its Comorbidities"

_ijms, 2024, doi:10.3390/ijms25020821_

Round 1
Reviewer 1 Report
Comments and Suggestions for Authors
Comment on manuscript N0: ijms-2771878
The authors in the review “Nrf2 pathway and oxidative stress as a common target for treatment of diabetes and its comorbidities” provide data on how diabetes increases oxidative stress, the general impact of that oxidative stress, and how oxidative stress primarily contributes to diabetic comorbidities, addressing how Nrf2 antioxidative pathway was affected, confirming the similar effects on the Nrf2 pathway of the heart, kidney, and liver systems.
Although the manuscript is well written several shortcomings should be corrected
When referring to plants, the first mention should give the full Latin name should be given, ‘species name, author name, and Family, while the species name should be in Italics (Aloe vera (L.) Burm.f., Asphodelaceae), afterward, it should be A. vera or common name. Please, check throughout the manuscript (Table 3, page 9). When addressing the products, the name should be precise, and in Italics, Genitive Ginkgonis extract, or if satisfying the Pharmacopoeia criteria Ginkgonis extractum siccum raffinatum et quantificatum (page 7).
Page 7, line 305, please, move the mentioned reference to the reference list section, and give the appropriate number, taking care of the changes that should be made for the rest of the references’ numbers.
Author Response
The authors in the review “Nrf2 pathway and oxidative stress as a common target for treatment of diabetes and its comorbidities” provide data on how diabetes increases oxidative stress, the general impact of that oxidative stress, and how oxidative stress primarily contributes to diabetic comorbidities, addressing how Nrf2 antioxidative pathway was affected, confirming the similar effects on the Nrf2 pathway of the heart, kidney, and liver systems. Although the manuscript is well written several shortcomings should be corrected
When referring to plants, the first mention should give the full Latin name should be given, ‘species name, author name, and Family, while the species name should be in Italics (Aloe vera (L.) Burm.f., Asphodelaceae), afterward, it should be A. vera or common name. Please, check throughout the manuscript (Table 3, page 9). When addressing the products, the name should be precise, and in Italics, Genitive Ginkgonis extract, or if satisfying the Pharmacopoeia criteria Ginkgonis extractum siccum raffinatum et quantificatum (page 7).
- Thank you for your suggestions. We gave the full Latin name of each plant mentioned in the manuscript: Cinnamomum verum Presl., Lauraceae (Line 197), Cinnamomum cassia (L.) J. Presl., Lauraceae (Line 204), Aloe vera (L.) Burm.f., Asphodelaceae (Line 412), Dracocephalum tanguticum Maxim., Lamiaceae (Line 522). All the plants were abbreviated appropriately after the first mention, to C. verum, C. cassia, A. vera, and D. tanguticum. Tables 1, 2, and 3 were also fixed with the appropriate plant names.
Page 7, line 305, please, move the mentioned reference to the reference list section, and give the appropriate number, taking care of the changes that should be made for the rest of the references’ numbers.
- Sorry for the confusion. It was moved to the reference section and given the reference number 100 in the text (Line 405).
Reviewer 2 Report
Comments and Suggestions for Authors
Inflammation appears to play a role in many conditions including diabetes, hypertension and other vascular disorders, renal, hepatic, cardiac, neurological and many others. The authors provide a thoughtful consideration for chronic condition and inflammation risk. They address and cite insightful literature on the genesis and causes of inflammation that may be unique or shared between many conditions. They describe and cite the type of inflammation in these chronic conditions and how it is distinct from infectious inflammation. They point out mechanisms of pathways that are potentially activated in many of these conditions and propose these as potential therapeutic intervention points, notably focusing on the Nrf2 pathway. It is adequately described and regulation of activation explained. The one section that should be enhanced are the agents that inhibit Nrf2. Apegenin is perhaps the most studied but luteolin,, chrysin and buterol have also been shown to be effective and some of these have been studied in animal models. There is a recent 2019 review on activators and inhibitors of Nrf2 https://www.ncbi.nlm.nih.gov/pmc/articles/PMC6664516/ that would be good to include. The section on studies that have been done is good and a few sentences on more what is known of Nrf2 inhibitor pharmacology is important.
Author Response
Inflammation appears to play a role in many conditions including diabetes, hypertension and other vascular disorders, renal, hepatic, cardiac, neurological and many others. The authors provide a thoughtful consideration for chronic condition and inflammation risk. They address and cite insightful literature on the genesis and causes of inflammation that may be unique or shared between many conditions. They describe and cite the type of inflammation in these chronic conditions and how it is distinct from infectious inflammation. They point out mechanisms of pathways that are potentially activated in many of these conditions and propose these as potential therapeutic intervention points, notably focusing on the Nrf2 pathway. It is adequately described and regulation of activation explained. The one section that should be enhanced are the agents that inhibit Nrf2. Apegenin is perhaps the most studied but luteolin,, chrysin and buterol have also been shown to be effective and some of these have been studied in animal models. There is a recent 2019 review on activators and inhibitors of Nrf2 https://www.ncbi.nlm.nih.gov/pmc/articles/PMC6664516/ that would be good to include. The section on studies that have been done is good and a few sentences on more what is known of Nrf2 inhibitor pharmacology is important.
- Thank you for a good suggestion. We added an explanation of some well-known Nrf2 inhibitors like luteolin, apigenin, wogonin, and chrysin (Lines 577-585).
Reviewer 3 Report
Comments and Suggestions for Authors
This paper describes in detail how drugs are explained to fight diabetes through signaling pathways via the Nrf2 pathway. The mechanisms of action of diabetes and its complications are also described in detail. However, the relationship between the Nrf2 pathway and oxidative stress needs to be elucidated in greater depth. The mechanism between how drugs act on complications through the Nrf2 pathway is not clear. Therefore, further revisions will be needed. Here are some comments.
1. In lines 53 to 104, is the hypothesis for the way in which diabetes causes oxidative stress confirmed? Especially Nrf2, please clarify in the paper.
2. Please add a diagram of the mechanism of how vascular hardening occurs through the Nrf2 pathway.
3. Please add more content to table 1, and please correct the format of table 1.
4. Please explain the novelty of the paper in the introduction section.
5. Regarding part 3.3, O-GlcNAcylation plays an important role in fighting cardiomyopathy. Please explain the link between O-GlcNAcylation and the Nrf2 pathway. And please explain why O-GlcNAcylation was depicted at great length.
6. Regarding lines 305 to 308, please check the references critically.
7. Please add a diagram about the process of detoxification of diabetic drugs through the Nrf2 pathway for a deeper understanding.
8. Regarding the conclusions section, there was not enough detail. Please explain in more detail the impact of the Nrf2/Keap1 pathway in diabetes and how it plays a role in complications arising in other organs. Please also explain in more detail the significance of this pathway and the purpose of the review.
Comments on the Quality of English LanguageMinor editing of English language required
Author Response
This paper describes in detail how drugs are explained to fight diabetes through signaling pathways via the Nrf2 pathway. The mechanisms of action of diabetes and its complications are also described in detail. However, the relationship between the Nrf2 pathway and oxidative stress needs to be elucidated in greater depth. The mechanism between how drugs act on complications through the Nrf2 pathway is not clear. Therefore, further revisions will be needed. Here are some comments.
- In lines 53 to 104, is the hypothesis for the way in which diabetes causes oxidative stress confirmed? Especially Nrf2, please clarify in the paper.
- Sorry for the confusion. We changed the term “hypotheses” to “mechanisms” to avoid confusion, since they have been confirmed with experiments (Line 60). Additional studies that corroborate each mechanism are discussed (Lines 73-77, 79-82, 90-92, 95-98).
- Please add a diagram of the mechanism of how vascular hardening occurs through the Nrf2 pathway.
- Thank you for your good suggestion. We added two diagrams: Figure 1 (the overall mechanism of the development of atherosclerosis caused by Nrf2 deficiency in arterial macrophages).
- Please add more content to table 1, and please correct the format of table 1.
- Thank you for your good suggestion. We added two more drugs/substances, liraglutide and verum/C. cassia (cinnamon), for atherosclerosis, and one more, sitagliptin, for cardiac fibrosis. They were explained in the text as well (liraglutide in Lines 184-196 and 248-256, C. verum/C. cassia in Lines 197-211 and 257-265, sitagliptin in Lines 321-330 and 371-380).
- Please explain the novelty of the paper in the introduction section.
- We added explanations that emphasize the novelty of the paper in the abstract (Lines 19-21) and the introduction (49-53).
- Regarding part 3.3, O-GlcNAcylation plays an important role in fighting cardiomyopathy. Please explain the link between O-GlcNAcylation and the Nrf2 pathway. And please explain why O-GlcNAcylation was depicted at great length.
- Thank you for your good suggestion. We explained the significance of O-GlcNAcylation in the development of diabetic cardiomyopathy (Lines 299-301). We also gave a thorough explanation of the link between O-GlcNAcylation and the Nrf2/Keap1 pathway (Lines 348-357).
- Regarding lines 305 to 308, please check the references critically.
- Sorry for confusion. It was moved to the reference section and given the reference number 100 in the text (Line 405).
- Please add a diagram about the process of detoxification of diabetic drugs through the Nrf2 pathway for a deeper understanding.
- Thank you for your good suggestion. We added Figure 2 (the mechanisms of antidiabetic drugs detoxifying the liver through the Nrf2/Keap1 signaling pathway).
- Regarding the conclusions section, there was not enough detail. Please explain in more detail the impact of the Nrf2/Keap1 pathway in diabetes and how it plays a role in complications arising in other organs. Please also explain in more detail the significance of this pathway and the purpose of the review.
- Thank you for your great suggestion. We added more detail about the Nrf2/Keap1 pathway and its significance (Lines 595-599, 602-604, and 609-611). The purpose of the review was also elaborated (Lines 604-608).
Round 2
Reviewer 3 Report
Comments and Suggestions for Authors
-